# Diagnostic ability of Peptidase S8 gene in the Arthrodermataceae causing dermatophytoses: A metadata analysis

**Apoorva R. Kenjar[1], Juliet Roshini Mohan Raj**[1]*, **Banavasi Shanmukha Girisha[2], Indrani Karunasagar[1]**

**1** Nitte (deemed to be University), Nitte University Centre for Science Education and Research, Mangaluru, Karnataka, India, **2** Nitte (deemed to be University), KS. Hedge Medical Academy, Mangaluru, Karnataka, India

* julietm@nitte.edu.in

**Data Availability Statement:** All relevant data are within the manuscript and its Supporting Information files.

## Abstract

An unambiguous identification of dermatophytes causing dermatophytoses is necessary for accurate clinical diagnosis and epidemiological implications. In the current taxonomy of the Arthrodermataceae, the etiological agents of dermatophytoses consist of seven genera and members of the genera *Trichophyton* are the most prevalent etiological agents at present. The genera *Trichophyton* consists of 16 species that are grouped as clades, but the species borderlines are not clearly delimited. The aim of the present study was to determine the discriminative power of subtilisin gene variants (SUB1-SUB12) in family Arthrodermataceae, particularly in *Trichophyton*. Partial and complete reads from 288 subtilisin gene sequences of 12 species were retrieved and a stringent filtering following two different approaches for analysis (probability of correct identification (PCI) and gene gap analysis) conducted to determine the uniqueness of the subtilisin gene subtypes. SUB1 with mean PCI value of 60% was the most suitable subtilisin subtype for specific detection of *T. rubrum* complex, however this subtype is not reported in members of *T. mentagrophytes* complex which is one of the most prevalent etiological agent at present. Hence, SUB7 with 40% PCI value was selected for testing its discriminative power in *Trichophyton* species. SUB7 specific PCR based detection of dermatophytes was tested for sensitivity and specificity. Sequences of SUB7 from 42 isolates and comparison with the ITS region showed that differences within the subtilisin gene can further be used to differentiate members of the *T. mentagrophytes* complex. Further, subtilisin cannot be used for the differentiation of *T. benhamiae* complex since all SUB subtypes show low PCI scores. Studies on the efficiency and limitations of the subtilisin gene as a diagnostic tool are currently limited. Our study provides information that will guide researchers in considering this gene for identifying dermatophytes causing dermatophytoses in human and animals.

**Funding:** The author(s) received no specific funding for this work.

**Competing interests:** The authors have declared that no competing interests exist.

## Introduction

Dermatophytes are a group of fungi that utilize keratinized tissue of humans and animals as a major nutritional source and hence cause infections that are cutaneous or restricted to the cornified layers such as skin, stratum corneum, hair and nails. Dermatophytoses is often painless and neglected until the appearance of large lesions, which later have remarkable effects on social and psychological quality of life [1]. The global burden of dermatophyte infection in human is estimated to be 20–25% [2]. However, in India, the prevalence of dermatophytoses could be 36.6–78.4% [3].

Accurate diagnosis is essential to effectively manage dermatophytoses. While based on direct microscopic examination of skin scrapings, the presence of fungal elements like conidia and true hyphae indicate fungal infections, the diagnosis is not conclusive and identification requires other culture-dependent or culture-independent methods. Recovery of cultures from samples is low due to the poor sensitivity and extended result turnaround time which can exceed four weeks [4]. The morphological similarities among closely related species complicate conventional identification and thus requires expertise and a panel of other tests for conclusive findings [5–7].

Molecular approaches have proven advantageous in resolving problems of laboratory diagnosis. Genotyping using methods like multiplex techniques, restriction digestion of the PCR products or real time detection targeting regions like the internal transcribed spacer region (ITS), chitin synthase encoding gene (CHS1) and large subunit rRNA gene have gained attention in recent years. Internal transcribed spacer region is the universal barcode for the identification of fungus [8]. However, the ITS sequencing alone cannot be used as a determinative tool in specific identification in case of some fungal taxa [9]. Chitin synthase, an enzyme involved in the synthesis of chitin, a major cell wall component of fungus, is also present in all fungi and hence cannot differentiate between dermatophytes and non-dermatophytes [10].

Protein markers have better species identification power [11] but are not well-studied in fungi. For organisms infecting skin, like the dermatophytes, secreted proteases play a significant role in establishing infection and proliferating in the host [12]. Martinez *et al* [13] reported a comparative genome analysis of *T. rubrum* and other related dermatophytes to identify candidate genes involved in infection. One of the genes selectively present and expressed in the dermatophytes is the Peptidase S8, also known as subtilisin. Secretome analysis of *T.behemiae* and *T.rubrum* indicate upregulation of the endoproteinases known as subtilisins and fungalysins that are commonly encountered in the dermatophytes. These enzymes digest proteins into large peptides and subsequently break them down into short chain peptides and amino acids. Preliminary sulfitolysis of native keratin is required for the action of several proteolytic enzymes. However, subtilisins are capable of degrading native keratin [14]. Thus, it was hypothesized that the subtilisin gene can be targeted for the specific identification of agents causing dermatophytoses in humans as well as animals.

The current taxonomy of the Arthrodermataceae groups the dermatophytes into seven genera. *Trichophyton*, the most prevalent etiological agent at present, further consists of 16 species that are grouped in four major complexes: the *Trichophyton mentagrophytes* complex, the *T. benhamiae* complex, *T.bullosum* and the *T.rubrum* complex wherein the species borderlines within each complex are not clearly delimited [15–17].

In this study, an extensive comparative analysis was performed using a dataset of all available subtilisin gene sequences for Arthrodermataceae to evaluate if this gene would be suitable for the unambiguous identification of the dermatophytes. Additionally, the subtilisin encoding gene subtypes in *Trichophyton* species, *Microsporum canis* and *Epidermophyton floccosum*

were investigated to determine the availability of these genes across species and the applicability for species differentiation among the dermatophytes.

## Materials and methods

### In-silico analysis

**Sequence acquisition, filtering and alignment.** All subtilisin (SUB1-12) sequences with taxonomic identifications deposited in NCBI were used in the study. Nucleotide FASTA sequence files were downloaded from NCBI nucleotide database (https://www.ncbi.nlm.nih.gov/nucleotide/), along with the accession numbers, fungal classification and information related to subtilisin gene subtypes.

The filters were manually applied to ensure that only sequences identified at the species level were retained. Species with 'sp.', 'unverified' names and nucleotide sequences less than 500bp were excluded. The subtilisin sequences were divided based on species and on expressed subtilisin subtypes (SUB1-SUB12). Sequences from each dataset were aligned using Multalin tool with default parameters [18].

To evaluate the differentiative power of subtilisin gene, Probability of Correct Identification (PCI) was determined using the pairwise p-distances generated using the distance matrix calculated in MEGA X (version 11.0.2) for interspecies and intraspecies differences with the default option. Minimum interspecific and maximum intraspecific pairwise distances were plotted as a box plot and examined for the presence of any overlap between the species [19, 20]. Two-sample unequal variances (heteroscedastic) test was performed to find the significant difference between the subtilisin subtypes. $P \leq 0.05$ was considered to be significant. All graphs were created using Excel (Microsoft Office 2021).

### In-vitro analysis

**Fungal isolates, growth conditions and DNA extraction.** *Trichophyton interdigitale* ATCC 9533, *Trichophyton rubrum* ATCC 28188, *Trichophyton tonsurans* ATCC 28942, and *Microsporum canis* ATCC 36299 (HiMedia India), anonymized clinical isolates from tinea corporis presentation, phenotypically identified (based on colony morphology, pigmentation, growth rate, and microscopic examination of hyphae, microconidia, macroconidia on lactophenol cotton blue mount preparation followed by hair perforation test and urease test) and genotypically based on sequencing of internal transcribed spacer region (data not provided) as *T. mentagrophytes* complex (n = 34), *Epidermophyton floccosum* (n = 1), *Nannizzia gypsea* (n = 3), saprophytes one each of *Rhizopus stolonifer*, *Aspergillus flavus*, and *A. niger* and nine *Penicillium* spp. isolated from the environment were included. Institutional ethics committee approval to use anonymized clinical isolates was obtained from Nitte Central Ethics Committee via sanction order NU/CEC/2019/0249 dated 14.08.2019.

Cultures were grown in Sabouraud's dextrose broth (SDB) at 28°C for 7 days and used for DNA extraction using a lyticase buffer-based extraction [21]. Briefly, cultures were treated with 500 μl of extraction buffer (100 mM Tris HCl, 25 mM EDTA, 1% β-mercaptoethanol, 10U lyticase), followed by lysis buffer (10 mM Tris, 10 mM KCl, 10 mM MgCl₂, 500 mM NaCl, 2 mM EDTA, 0.5% SDS) and proteinase K (20 mg/ml). Following incubation, extraction with phenol-chloroform isoamyl alcohol (25:24:1) and precipitation with isopropanol. The alcohol precipitated DNA was recovered by centrifugation and the pellet dissolved in 30μl of nuclease-free water.

**Optimization of subtilisin gene amplification, sequencing and analysis.** Oligonucleotide primers used for polymerase chain reaction were designed based on the complete subtilisin (SUB7) gene sequence representing the Arthrodermataceae (KF146903, FJ348241,

AY439111, AY437852). The specificities of the designed primers were initially checked using the Basic Local Alignment Search Tool (BLAST) program on the National Center for Biotechnology Information server (http://www.ncbi.nlm.nih.gov/blast) before being confirmed by the assays. Reactions were carried out in 30μl volume containing 3μl of 10X buffer (100 mM Tris-HCl, 1.5 mM MgCl$_2$), 1μl containing 2.5 nM concentrations of each deoxyribonucleotide phosphates, 1μl containing 10 pM of primer, 1U of Taq DNA polymerase (HiMedia Laboratories Pvt. Ltd., India), 100ng of template DNA and remainder made up with nuclease free water. The reactions were carried out using a thermal cycler (Eppendorf Nexus G2) with initial denaturation at 95˚C for 5 minutes, 35 cycles of 95˚C for 30 seconds, 42˚C annealing temperature for 30 seconds and 72˚C for 30 seconds followed by a final extension at 72˚C for 10 minutes. All the products of the thermocycling reaction were visualized by electrophoresis on 2% agarose gel stained with Hi-SYBR safe (HiMedia Laboratories, India) using a UV transilluminator with documentation system (BioRad, CA, USA).

PCR amplification products were outsourced for Sanger sequencing (Eurofins Genomics Pvt Ltd, Bengaluru, India). The consensus nucleic acid sequence obtained from forward and reverse primer cycle sequences were trimmed using Chromas software and then aligned using NCBI BLAST and Expasy translate (https://web.expasy.org/translate/). Multiple sequence alignment of the nucleotide sequences was performed using the Multalin tool and Clustal-Omega (https://www.ebi.ac.uk/jdispatcher/msa/clustalo). Gene similarity was determined from the best scoring reference sequence from NCBI, similarity output with an identity of >95% and >98% coverage was considered. The sequences were submitted to Genbank (NCBI). Association of different genera was based on tree-based analysis and phylogenetic trees were constructed using Neighbour Joining method in MEGA X (version 11.0.2) with bootstrap tested for 1000 replicates. To expand the spectrum, subtilisin sequences of standard cultures from NCBI nucleotide were also included.

## Results

### In-silico analysis

**Sequence acquisition, filtering and alignment.** Primary dataset comprising of all subtilisin sequences belonging to Arthrodermataceae consisted of 288 sequences. The exclusion of sequences without species names, subtilisin subtypes and sequences with less than 500 bp resulted in 237 sequences. This dataset was used to perform all subsequent analyses as represented in Fig 1A where the data is divided based on 12 species including *Trichophyton rubrum*, *T. mentagrophytes*, *T. benhamiae*, *T. tonsurans*, *T. interdigitale*, *T. verrucosum*, *T. soudanense*, *T. equinum*, *T. violaceum*, *Microsporum canis*, *Epidermophyton floccosum*, and *Arthroderma uncinatum*. The number of sequences in our trimmed dataset is as follows: 94% (223 sequences) belong to species of *Trichophyton*, only 3.7% (9 sequences) from *Microsporum*, 1.2%. (3 sequences) from *Epidermophyton* and 0.8%. (2 sequences) from *Arthroderma*. In Fig 1B, the data is divided based on subtilisin subtypes (SUB1-12). The number of sequences in our trimmed dataset is as follows: 20.3% (48 sequences) belong to subtype SUB5, 16.5% (39 sequences) to subtype SUB3, 14.3% (34 sequences) to each subtype SUB1 and SUB4, 13.9% (33 sequences) belong to subtype SUB2, 10.1% (24 sequences) belong to subtype SUB6, 7.6% (18 sequences) belong to subtype SUB7, 1.3% (3 sequences) belong to subtype SUB11, 0.4% (1 sequence) belong to each subtype SUB8, SUB9, SUB10, and SUB12 (Fig 1B). The list of species included to perform the analysis is provided with accession number and type of expressed subtilisin (S1 Table).

The probability of correct identification (PCI) was analyzed for seven subtilisin subtypes (SUB1-7) due to the unavailability of sufficient sequences in other subtypes (SUB8-12) in most

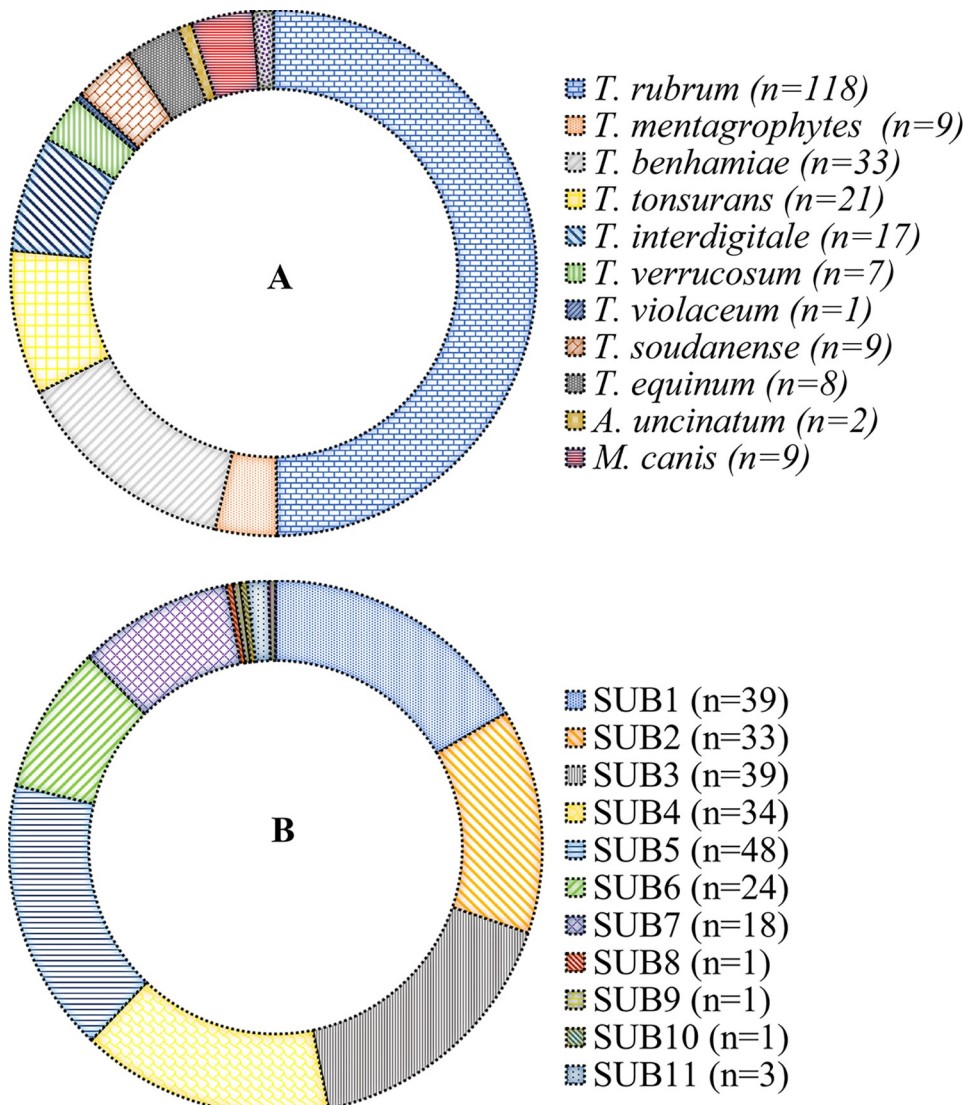

**Fig 1.** Pie charts representing the numbers of sequences available for A. each species B. each subtilisin subtypes in the dataset used in the study.

of the species. Thus, the number of sequences and species analyzed decreased to 233 sequences and 10 species. SUB1 showed good probability of identification for *T.rubrum*, *T.benhamiae* and *M.canis* with the coverage of 88%, followed by *T.interdigitale* and *T.soudanense* with 75% and *T.tonsurans*, *T.verrucosum* and *T.equinum* with 63%. SUB3 showed 60% for *T.interdigitale* and *T.soudanense*, 50% to *T.rubrum* and all other species showed less than 50%. SUB4 showed highest identification percentage for *T.equinum* (86%), *T.tonsurans* (71%), *T.interdigitale* (57%) and *T.verrucosum* (57%). SUB6 showed 50% for *T.rubrum*, *T.interdigitale*, *T.souda-nense*, *T.equinum* and *T.mentagrophytes*. SUB7 showed 63% for *T.interdigitale*, 50% each for *T.rubrum*, *T.benhamiae*, *T.tonsurans*, *T.soudanense*, *T.equinum* and *T.mentagrophytes*. Two subtilisin subtypes, SUB2 and SUB5 presented low (0–9%) probability of identification. The probability of correct identification value expressed in percentage for all the subtilisin subtypes are represented in Fig 2A and 2B.

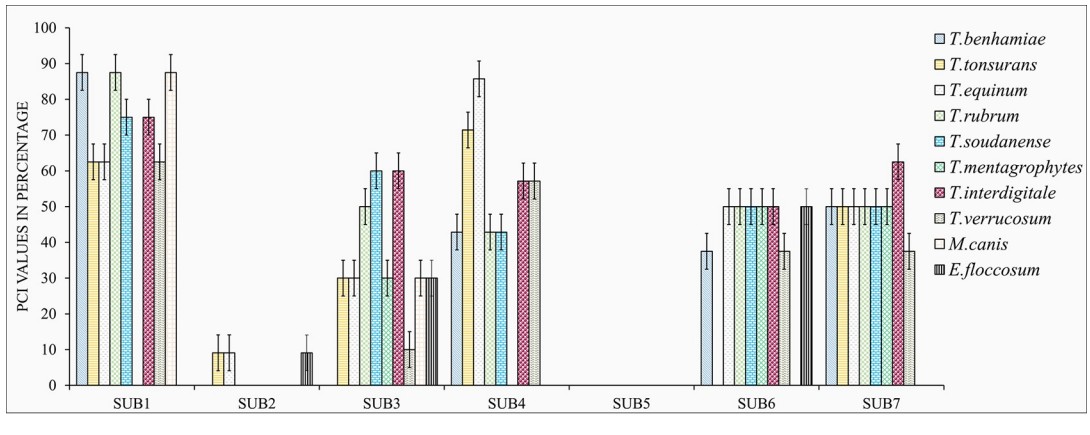

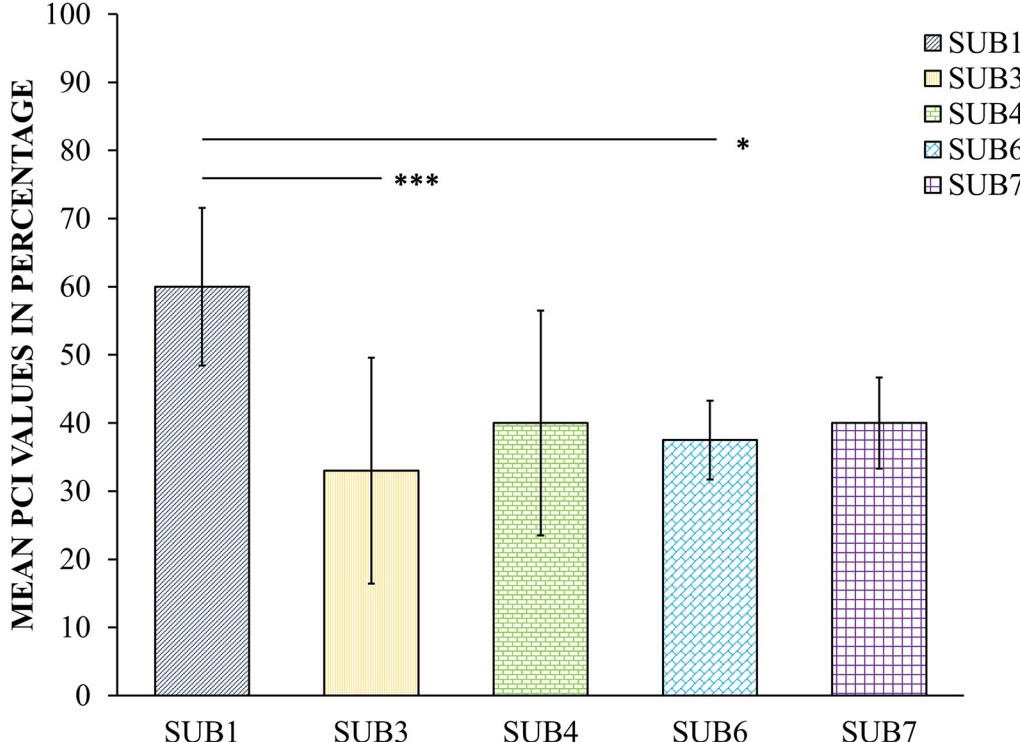

**Fig 2. Probable correct identification values for all of the Arthrodermataceae genera from trimmed dataset.** A. The PCI values were estimated for all subtilisin subtypes for each species based on sequence similarity. B. Comparison between subtilisin subtypes (SUB1-SUB7). Significant difference between the subtypes is represented as * ($p \leq .05$), ** ($p \leq .01$), *** ($p \leq .001$). SUB2 and SUB5 are eliminated since the PCI score is <10%.

Two-sample unequal variances t-test was used to compare subtilisin subtypes PCI scores. SUB1 versus SUB4 and SUB7, SUB3 versus SUB4, SUB6 and SUB7, SUB4 versus SUB6 and SUB7, SUB6 versus SUB7 subtypes comparisons were not statistically significant indicating that the subtypes SUB1, SUB3, SUB4, SUB6 and SUB7 could be used for the identification of dermatophytes (Fig 2B) (S2 Table).

The sequence dataset was analyzed for gene gap between the intraspecies and interspecies of subtypes to assess and compare the efficiency of each subtilisin gene subtype in the identification of dermatophyte species. Defined gap between the intraspecies and interspecies including the whiskers states good identification power, even if outliers were overlapping. If the

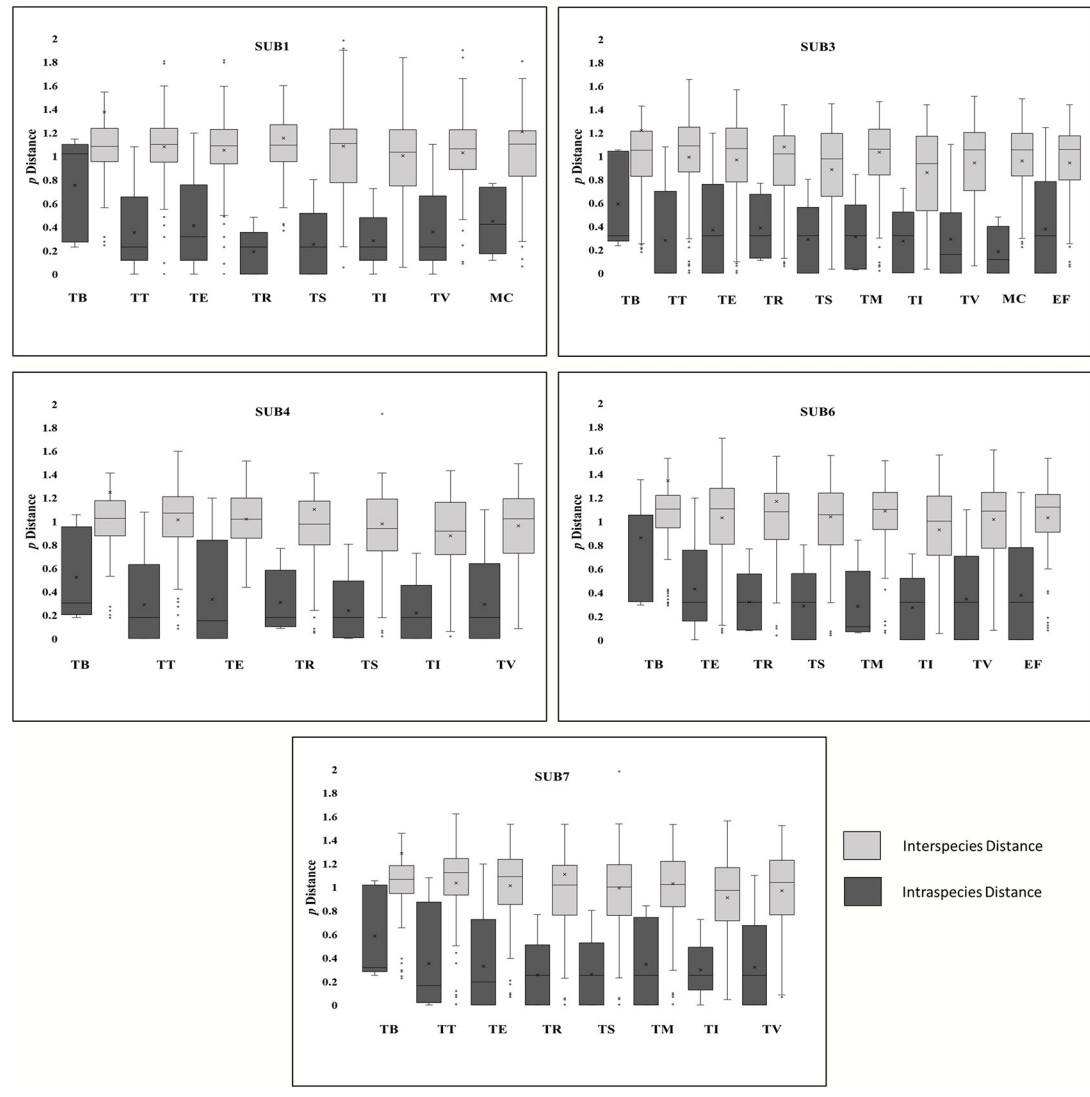

**Fig 3. Box plot plotted for intraspecies and interspecies distance to demonstrate the subtilisin gene overlap for a given species.** TB: *T.benhamiae;* TT: *T.tonsurans; TE*: *T.equinum;* TR: *T.rubrum*; TM: TS: *T.soudanense; T.mentagrophytes*; TI: *T. interdigitale;* MC: *M.canis;* TV: *T.verrucosum;* EF: *E.floccosum.* A: SUB1; B: SUB3; C: SUB4; D: SUB6; E: SUB7.

whiskers from an intraspecies distance overlap interspecies distance it has intermediate power and poor identification if the boxes overlap. Most of the subtilisin subtypes were classified as intermediate for respective species while SUB1 has superior power to identify *T.rubrum* (Fig 3A–3E).

## In-vitro analysis

Though SUB1 and SUB4 show better identification score in identifying most of the *Trichophyton* species, sequences are not available for one of the most common causative members of the *Trichophyton mentagrophytes* complex i.e *Trichophyton mentagrophytes*. SUB 3 showed <50% identification score for five *Trichophyton* species (*T. benhamiae*, *T.tonsurans*, *T.equinum*, *T. mentagrophytes*, *T.verrucosum*) though the identification score for *T. rubrum*, *T.soudanense* and *T.interdigitale* was good. SUB 6 sequences are not available for *T.tonsurans*, additionally,

**Table 1. Dermatophyte subtilisin specific primer, sequences and annealing temperatures.**

| Primer name | Primer sequence | Predicted annealing temperature ˚C |
|---|---|---|
| SUBF | TACATTGTTGTCATGAA | 42 |
| SUBR | ATACCCATGATGACAC | 45 |

identification score for *T. benhamiae* and *T.verrucosum* were <50%. Thus, the in-vitro analysis was focused on SUB7 which was present in most of the *Trichophyton* species and expressed better identification score (50%).

## Amplification of SUB7, sequencing and phylogenetic analysis

Oligonucleotide primers used for polymerase chain reaction are listed in Table 1. The PCR amplification for SUB7 yielded ~650 bp product for *Trichophyton, Microsporum, Nannizzia* and *Epidermophyton* species tested. The negative controls: *Rhizopus stolonifer, A. flavus, A. niger* were negative while *Penicillium* species. showed non-specific amplification with multiple amplicon sizes varying from 100bp to 1000 bp with different species (Fig 4 and S1 Fig). The subtilisin reads from the anonymised *T. mentagrophytes* complex clinical isolates showed eight reads identical to *T. mentagrophytes* (>90% identity) and 26 isolates similar to *T.tonsurans* with an additional 54 bp nucleotide which is absent in *T. mentagrophytes*. (>90% identity). The subtilisin sequences of *N. gypsea* were identical to the standard *N. gypsea* subtilisin sequence available in GenBank. The subtilisin reads of 4 standard strains and 22 isolates were submitted to NCBI nucleotide database and accession numbers assigned (Table 2) Reads of remaining 16 isolates are provided in S1 File. The phylogenetic tree for subtilisin produced four groups. Group I clustered with the reference sequence for *T. mentagrophyte*. Group II clustered with the reference sequence *T. interdigitale, T.rubrum*, and non-*Trichophyton* species. Group III clustered with the reference sequence for *T. tonsurans*. Group IV cluster comprised of *T.mentagrophytes* complex isolates. (Fig 5). To further determine the specificity of subtilisin primers, all the translated Arthrodermataceae subtilisin sequences were aligned with the reads available for *Penicillium* alkaline serine proteases and the uncharacterized protein of Penicillium (S2 Fig). Most of the *Penicillium* alkaline serine proteases cluster as a separate branch while *Penicillium chrysogeneum* protein clusters with SUB 8, *P.waksmanii* protein clusters as a separate branch within SUB 2, *P citrinum* uncharacterized protein with *T. equinum* SUB 6 and *T.verrucosum* SUB 1. The amplification products obtained on thermocycling with the primers designed in this study for the *Penicillium* species were identical to the *P.citrinum* uncharacterized protein (XM_056639834) (S2 File). This protein shows less than 50% coverage i.e only 150bp of ~1200bp identical with complete subtilisin gene of *Trichophyton*.

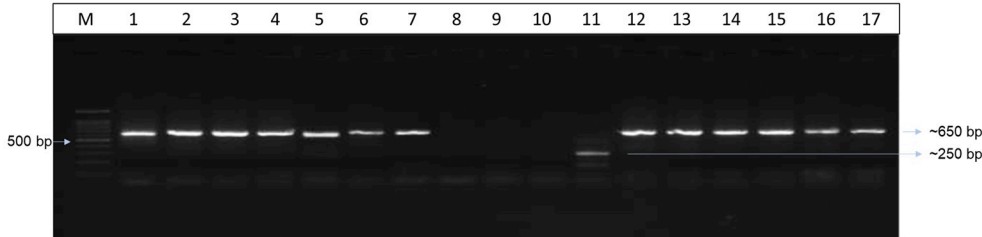

**Fig 4. PCR targeting subtilisin region.** M:100 base pair marker; 1: *Trichophyton interdigitale*; 2: *T. mentagrophytes*; 3: *T. rubrum*, 4: *T. tonsurans*; 5: *M. canis*; 6: *N. gypsea*; 7: *E. floccosum*; 8: *Rhizopus stolonifer.*; 9: *Aspergillus niger*; 10: *A. flavus*; 11: *Penicillium citrinum.*; 12–17: *Trichophyton* Clinical isolates.

**Table 2. Accession numbers obtained for NCBI submissions.**

| Sl no | Organism | GenBank Accession ID for Subtilisin region |
|---|---|---|
| 1 | ATCC 9533 *T. interdigitale* | OM320817 |
| 2 | *ATCC 28188 T. rubrum* | MW149250 |
| 3 | *ATCC 28942 T. tonsurans* | MW149247 |
| 4 | *ATCC 36299 M. canis* | MW149248 |
| 5 | *T. mentagrophytes complex* sample1 | MW149241 |
| 6 | *T. mentagrophytes complex* sample2 | MT711980 |
| 7 | *T. mentagrophytes complex* sample3 | MW149242 |
| 8 | *T. mentagrophytes complex* sample4 | MW149243 |
| 9 | *T. mentagrophytes complex* sample5 | OM397953 |
| 10 | *T. mentagrophytes complex* sample6 | OM397954 |
| 11 | *T. mentagrophytes complex* sample7 | OM397955 |
| 12 | *N.gypsea* sample8 | OM397964 |
| 13 | *T. mentagrophytes complex* sample9 | OM397956 |
| 14 | *T. mentagrophytes complex* sample10 | OM397957 |
| 15 | *T. mentagrophytes complex* sample11 | OM397958 |
| 16 | *T. mentagrophytes complex* sample12 | OM397959 |
| 17 | *T. mentagrophytes complex* sample13 | OM397960 |
| 18 | *N.gypsea* sample14 | OM397965 |
| 19 | *T. mentagrophytes complex* sample15 | MW149244 |
| 20 | *T. mentagrophytes complex* sample16 | OM397961 |
| 21 | *T. mentagrophytes complex* sample17 | OM397962 |
| 22 | *T. mentagrophytes complex* sample18 | OM397963 |
| 23 | *T. mentagrophytes complex* sample19 | MW149245 |
| 24 | *T. mentagrophytes complex* sample20 | MW149246 |
| 25 | *N.gypsea* sample 21 | MW149249 |
| 26 | *E. floccosum* sample22 | OM333895 |

## Discussion

Several microscopy dependent techniques play a vital role in the direct detection of fungal elements by wet mount preparation or stained slide preparation using bright field, phase contrast and fluorescence microscopy for the laboratory diagnosis of dermatophytoses. Though these techniques indicate a fungal infection, they are highly nonspecific [22–27]. The isolation and identification of etiological agent of dermatophytoses has long depended on culturing methods which stand as gold standard with the development of several modifications in SDA and potato dextrose agar with the inclusion of antibiotics widely used [28–30]. Major drawback of this gold standard is the low recovery of cultures from clinical specimen (<40%) and the long turnover time [31, 32].

Molecular diagnostics platforms provide rapid detection and better sensitivity. Matrix-assisted laser desorption/ionization-time of flight mass spectrometry (MALDI-TOF) which scans for surface biomolecules is currently used for the rapid identification of a wide range of bacteria and yeast species. However, its application for dermatophyte detection is currently limited to the availability of custom databases [33–35]. Studies highlight the misidentification of *Trichophyton* species due to insufficient spectra database available for dermatophyte identification [36, 37].

Molecular diagnosis for the dermatophytes is often by nucleic acid amplification techniques targeting the ITS and CHS1 regions [32, 38–40]. Several modifications of PCR like multiplex

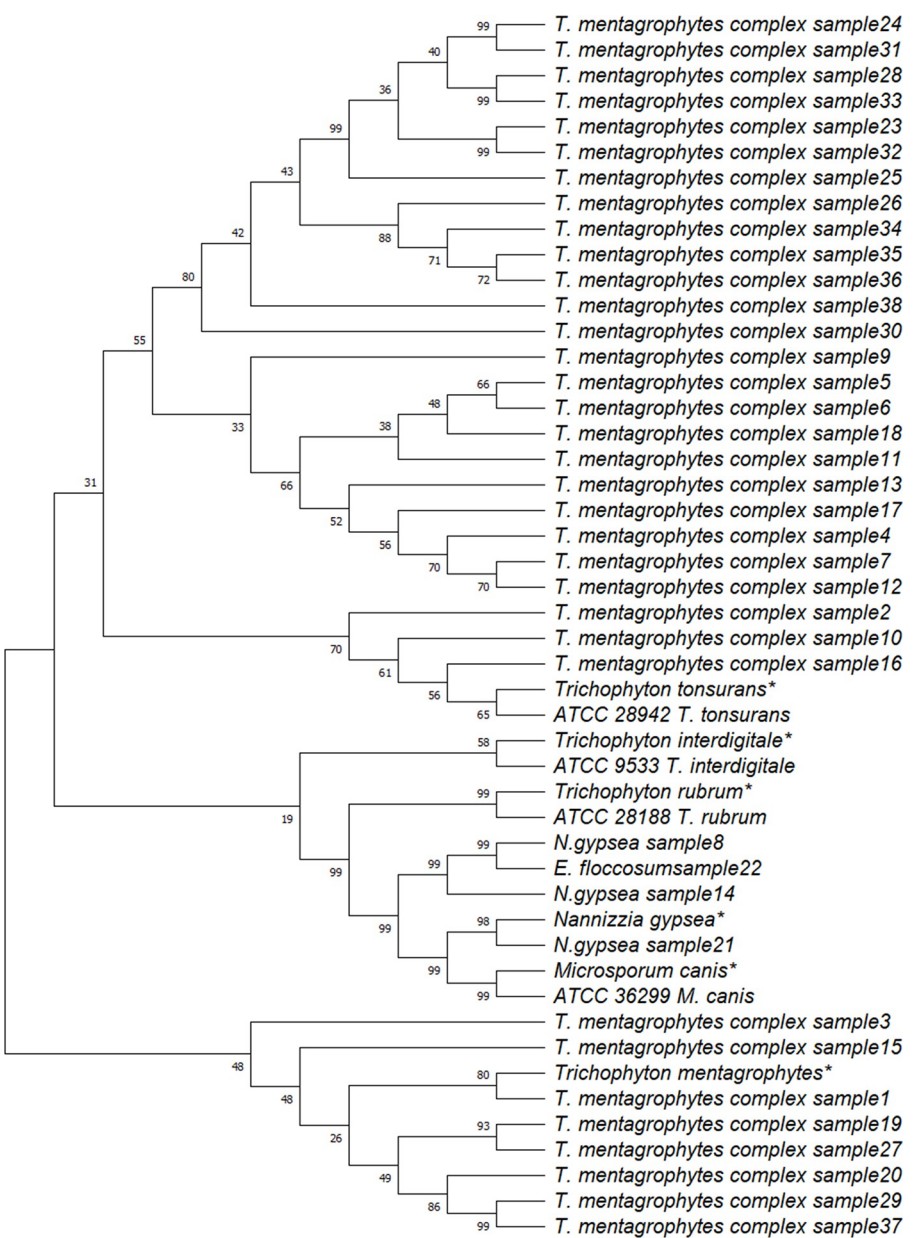

**Fig 5. Phylogenetic tree for diversity of species based on subtilisin region obtained in the study: Eight isolates identified as _T. mentagrophytes_ clustered with standard _T. mentagrophytes_ reference sequence, 26 isolates clustered as _T. mentagrophyte_ complex.** Three isolates clustered with _T. tonsurans_ reference sequence. The three isolates identified as _N. gypsea_ and one isolate identified as _Epidermophyton floccosum_ clustered around reference sequences of _M. canis_ and _N. gypsea_. Sequences marked * indicate that these sequences were taken from NCBI nucleotide. All other sequences were generated in this study.

techniques [41–43], nested PCR [44], PCR-ELISA [45], restriction digestion of the PCR products [46, 47] or real time detection [48, 49] for these targets are reported. The use of ITS and chitinase based molecular detection methods have proven valuable [50]. However, the universal presence of these genes hinders specific detection and identification [51]. ITS region is successful in identifying organisms belonging to phylum Basidiomycota whereas for organisms belonging to phylum Ascomycota the success rate for the identification is reported lower [52].

Some organisms like *Aspergillus* have ITS region identical to other clinically important species [53]. The universal presence of ITS region in all eukaryotic organisms does not help differentiate dermatophytes from non-dermatophytes and within the dermatophytes. Thus, the correct identification of the dermatophytes based on ITS sequencing remains disputed [17,54, 55].

Dermatophytes have the capacity to directly penetrate the cuticle using a large pool of hydrolytic enzymes among which proteases, specifically subtilisin-like serine proteases, are recognized as virulence factors [56, 57]. Despite the fact that the dermatophyte-host relationship is still not fully understood, the secretion of proteases is known to affect host defense processes along with breaking down proteins to provide nutrients for dermatophytes [58–60]. Jousson *et al* stated that subtilisin gene including SUB1, SUB3–SUB7 are specific to dermatophytes and appear to have emerged through successive gene duplication events [61]. Several studies have isolated, sequenced, characterized and decoded subtilisin gene SUB1-7 from *Trichophyton tonsurans* [62], *Microsporum canis* [63], *Trichophyton rubrum* [64–67], *Epidermophyton floccosum* [68], *Trichophyton mentagrophytes* and *Trichophyton benhamiae* (earlier *Arthroderma benhamiae*) [69, 70] in order to better understand the virulence mechanisms of these fungi. The current understanding of the subtilisin gene in dermatophytes is poor, however, the application of PCR for subtilisin in diagnosis has not been previously evaluated. In the present study we evaluated the ability of subtilisin gene for identification of dermatophytes using distance matrix analysis and sequencing SUB7 subtype of subtilisin. All subtilisin sequences of species belonging to the Arthrodermataceae from GenBank were filtered to obtain the most reliable results with the aim of creating high quality data output that would meet the theoretical assumptions of the biological system of identification. The number of sequences available for each species varies due to which there could be discrepancy in the analysis. However, with the manual filtering of incomplete sequences, we tried to reduce discrimination between the species which will in turn produce accurate results. Ninety-three percentage of the sequences belonged to *Trichophyton* species which comprise the most common causative agents of dermatophytoses in humans and animals. Based on PCI determination, SUB2 and SUB5 had least PCI rate which indicates that these regions are not suitable for identification of dermatophytes, thus these subtypes were not included for further analysis. SUB3 and SUB6 exhibited less than 50% PCI values for most of the species. SUB1 and SUB4 were the best regions for identification of most of the dermatophytes including *T.benhamiae*, *T.rubrum*, *M.canis*, *T. soudanense* and *T.interdigitale*, *T.tonsurans* and *T.equinum* however, sequences of SUB1 and SUB4 is not available for leading causative agent, *T.mentagrophytes*. Interspecies and intraspecies gap analysis of SUB1 and SUB4 indicate these genes are able to differentiate the members of the *T.rubrum* complex i.e *T.rubrum*, *T. soudanense* and *T. violaceum*. Sequence gap analysis for overlap showed that the five selected subtypes were intermediate for identification of species, except in the case of *T. benhamiae* wherein there was no significant sequence gap at both intraspecies and interspecies levels. *T.benhamiae* complex members are zoophilic and hence the proteases available in these species may differ from isolates that are more anthropophilic. Hence SUB7 with 50% PCI power for most of the species was chosen.

In the present study it is interesting to note that all the targeted dermatophytes showed amplification with the designed primers, which is contradiction to the study conducted by Khedmati *et al* [68] where the *Microsporum* and *Epidermophyton* were found to lack SUB4–7 genes. Among 34 isolates which were identified as *Trichophyton mentagrophytes* complex based on phenotype and ITS region analysis, 8 isolates had SUB7 sequences identical to *Trichophyton mentagrophytes*, the other 26 isolates showed similarities with the other *T. tonsurans* and *T. interdigitale* species which possessed additional 57 bp nucleotide that is absent in *Trichophyton mentagrophytes*. Additionally, recent study conducted by Kumar *et al* stated that the overall architecture of the genomes of *T. mentagrophytes* genotype VIII or *T. indotineae* were found to be similar to that of *T. interdigitale* strain with no major difference in the

predicted gene families involved in virulence and infection [71]. However, the SUB7 gene needs to be extensively explored in order understand the status of this gene in *T. mentagrophytes*, *T. indotineae* and *T. interdigitale* to determine its applicability in further differentiating the debatable members of the *T. mentagrophytes* complex. *T. indotineae* has emerged as a notable pathogen due to high terbinafine resistance associated with this species and its global presence. However, the identification of this species currently relies on genome sequencing [54,72, 73] and subtilisin sequences for this species are not available in databases. The additional amino acid residues reported in the present study may be associated with the severity of the infections caused by these isolates which needs to be co-related with antimicrobial susceptibility testing and clinical manifestations.

The specificity and sensitivity of a diagnostic method contributes to its application. The primers designed in this study aimed to specifically target the SUB7 of the dermatophytes. However, a similar protein is present in some species of *Penicillium* that is yet to be completely characterized. Nonetheless, the ability of the designed primers to differentiate dermatophytes from non-dermatophytes has been demonstrated. Studies report that proteases and subtilisin like genes are present in fungal saprophytes, phyto-pathogens, insect pathogens and human pathogens. Changes in the amounts and type of secreted proteases could be due to ecological switching related to a differential expression rather than genetic divergences of the genes encoding orthologous proteins [74, 75].

A major setback in culture dependent methods of fungal pathogens is the sensitivity and turnaround time. Rapid immunological test methods like ELISA and latex agglutination tests are proven sensitive for bacterial and viral infections, however, the sensitivity in fungal infections is low [76, 77]. Despite enhanced sensitivity in terms of amount of template required in nucleic acid-based detection methods like qPCR, ITS and chitin targeting methods will continue to be non-specific due to their universal presence in all species of fungi. The extraction of fungal nucleic acid from the clinical specimen is an additional challenge. Non-nucleic acid-based identification technologies of spectral analysis like nano- electrospray ionization mass spectrometry to detect and differentiate the secreted proteins like subtilisin with turnover time within 10 minutes, with a requirement of less than 50 pmol of a substance and creation of superior databases for techniques like MALDI-TOF or applications in nanoparticle enhanced ELISA can revolutionize dermatophyte diagnosis.

## Conclusion

Clinical and epidemiological areas critically depend on the rapid and accurate identification of microbial species. Thus, studies related to identification of species are crucial. Knowledge regarding the efficiency and limitations of the protein markers that are currently used for specific organisms are limited. The present study contributes to the identification of species belonging to the family Arthrodermataceae targeting subtilisin gene, a protein marker. According to literature, the subtilisin gene studied is more specific for the detection of dermatophytes. However, the gene has not been exploited in order to be used for identifying species. Our study provide knowledge that could help scientists to select the best protein markers for identification and classification of dermatophytes which will interim be useful to start early therapeutic measures in clinical setting in cases of dermatophytoses in humans as well as animals.

## Supporting information

**S1 Table. The list of species included in this study with nucleotide accession number and type of expressed subtilisin.**
(PDF)

**S2 Table. Estimation of statistical significance in the computed phylogenetic distance (*p* value) of different subtilisin subtypes.**
(PDF)

**S1 Fig. PCR targeting ITS region and Subtilisin region for *Penicillium* species isolated from environment.**
(PDF)

**S2 Fig. Phylogenetic tree for the alkaline protease/uncharacterized protein of *Penicillium* species available in NCBI in comparison with the subtilisin of Arthrodermataceae species.**
(PDF)

**S1 File. Sequences of 16 isolates used for phylogenetic tree construct (Fig 5) which are not uploaded to NCBI nucleotide database.**
(PDF)

**S2 File. Sequences obtained from amplification of *Penicillium* species using the primers designed in this study.**
(PDF)

## Author Contributions

**Conceptualization:** Juliet Roshini Mohan Raj, Indrani Karunasagar.

**Data curation:** Apoorva R. Kenjar.

**Formal analysis:** Juliet Roshini Mohan Raj.

**Investigation:** Apoorva R. Kenjar.

**Methodology:** Apoorva R. Kenjar.

**Resources:** Banavasi Shanmukha Girisha.

**Validation:** Juliet Roshini Mohan Raj.

**Visualization:** Apoorva R. Kenjar.

**Writing – original draft:** Apoorva R. Kenjar.

**Writing – review & editing:** Juliet Roshini Mohan Raj, Banavasi Shanmukha Girisha, Indrani Karunasagar.

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
