## [Decision Letter · Decision Letter 0]

9 Apr 2024

PONE-D-24-06282Diagnostic ability of Peptidase S8 gene in the Arthrodermataceae causing dermatophytosis: A Metadata analysisPLOS ONE

Dear Dr. Mohan Raj,

Thank you for submitting your manuscript to PLOS ONE. After careful consideration, we feel that it has merit but does not fully meet PLOS ONE’s publication criteria as it currently stands. Therefore, we invite you to submit a revised version of the manuscript that addresses the points raised during the review process. In particular, the selected primer pairs need to be more extensively evaluated for their specificity and more strains need to be tested. The chosen alignment temperature is relatively low and the selected primer sequences show very high sequence similarity with alkaline serine proteases of Penicillium species. Are the authors sure that the amplified 200 bp with Penicillium DNA is unspecific? This could be checked by sequencing. I also reviewed the primer sequences with BLAST and found e.g. 100% sequence similarity of the FW primer with Penicillium verrucosum (XM_057218044) and Penicillium chrysogenum (AJ870492.2). Also the RW primer shows high sequence similarities with some Penicillium species. Therefore several Penicillium species should be tested by the authors. Does the amplified sequence always differ in size? Do the primers target the alkaline serein protease sequences of Penicillium species? These questions should be addressed in a revised manuscript. Furthermore, the introduction needs some brief information about the subtilisin genes and the discussion has to include a comparison of the author's method with already established methods for the identification of Arthrodermataceae species. What are the pros and cons?

We look forward to receiving your revised manuscript.

Kind regards,

Olaf Kniemeyer

Academic Editor

PLOS ONE

Journal Requirements:

Reviewers' comments:

Reviewer's Responses to Questions

**Comments to the Author**

1. Is the manuscript technically sound, and do the data support the conclusions?

Reviewer #1: Yes

Reviewer #2: Partly

2. Has the statistical analysis been performed appropriately and rigorously? 

Reviewer #1: Yes

Reviewer #2: Yes

3. Have the authors made all data underlying the findings in their manuscript fully available?

Reviewer #1: Yes

Reviewer #2: Yes

4. Is the manuscript presented in an intelligible fashion and written in standard English?

Reviewer #1: Yes

Reviewer #2: Yes

5. Review Comments to the Author

Reviewer #1: Review for manuscript assigned as: PONE-D-24-06282 (Manuscript Number)

Dear Authors,

in my humble opinion your work is very interesting and has not only cognitive, but also applied character. This paper contributes a lot to medical mycology, molecular diagnostics of fungi, epidemiology, and even taxonomy and phylogenesis of dermatophytes. Issues described by Authors allow to show new possibilities in the fast diagnostics of dermatophytoses. The Authors put a lot of work to perform in silico analyses. Nevertheless, as a reviewer, I am obliged to point out major and minor shortcomings in the work.

So in my opinion, Introduction lacks basic information about subtilisin. It would be good to at least briefly mention what subtilisins are and what is their biological role. I would see here a few sentences like "Subtilisin-like proteases are a group of serine endoproteases, coded by seven genes (SUB1-7), which decompose keratin structures." It is worth emphasizing in your article why it is worth dealing with such proteins and encoding them genes. Such information will certainly be helpful to potential readers and will attract more interested researchers, and will also increase the attractiveness of your article. Additionally, in the "Discussion" the Authors should refer to other molecular methods used in the detection of dermatophyte species, such as matrix-assisted laser desorption/ionization time of flight mass spectrometry (MALDI-TOF MS) and nano-electrospray ionization mass spectrometry (nano -ESI-MS) to detect and identify the most frequently isolated dermatophytes in comparison with subtilisin gene used as protein marker. How and why would the solution based on sequences for subtilisins proposed by Authors be better?

All the figures and tables are appropriate for this type of article but require thorough correction. Figures in their current form are unacceptable. In general, the paper has a logical flow. The abstract well correspond with the main aspects of the work and the literature is properly selected.

As a reviewer I am obligated to pay attention even to less important weak points of this work and all mentioned below comments should be carefully considered.

Title

I would suggest changing the title to a more appropriate one, namely: “The diagnostic potential of Peptidase S8 gene in the Arthrodermataceae causing dermatophytoses: A Metadata analysis”. Moreover, due to the fact that "Arthrodermataceae" are in the plural, I also propose changing ,,dermatophytosis” to "dermatophytoses"

Abstract

Due to the repetition of the term "accurate", I suggest rephrasing the first sentence to more appropriate, namely "Unambiguous identification of dermatophytes is necessary for accurate clinical diagnosis and epidemiological implications".

As I know should be ,,...of T.mentagrophytes species complex...”

In my opinion ,,...dermatophytes causing dermatophytoses..." sounds more correct.

To the best of my knowledge the genus Trichophyton split into three separate clades, namely: T. rubrum complex, T. benhamiae complex and T. mentagrophytes complex. The Authors did not mention anything about T. rubrum complex in the abstract. Can subtilisin be used for the differentiation of T. rubrum and what about PCI scores for several subtypes? In my opinion it will be significant for potential readers to know if differences within the subtilisin gene can further be used to differentiate members of the T. rubrum complex. However, if the Authors have not conducted such research, it is worth informing about it at least briefly. While all subtypes of subtilisin (gene variants) show low PCI scores and for this reason cannot be used for the differentiation of T. benhamiae species complex, SUB7 seems to be the best for T. mentagrophytes, but which one should be used for the differentiation of T. rubrum species complex? The Authors do mention it in "Results" (line 105), I quote "SUB1 showed good probability of identification for T. rubrum" but I think it is worth mentioning T. rubrum complex in the Abstract for full clarity.

Introduction

Line 2

As I know should be ,,... humans and animals...”

Line 3

I can`t agree that skin is ,,the non-living cornified layer”. Skin should be considered as an organ.

Line 6

The article by Kakande et all [2] refers to the prevalence of dermatophytoses in Uganda. Instead of this article I would like to suggest to cite more appropriate work in this field (for example: DOI:https://doi.org/10.1016/j.pathol.2016.08.006)

Line 9

To the best of my knowledge should be "While direct microscopic examination of skin scrapings indicates fungal infection, ...". Based on microscopic examination (10%KOH in DMSO or 5mM calcofluor white) we can observe blastoconidia, true hyphae and other fungal structures depending on fungal etiological agent of infection. If true hyphae is observed it may be dermatophytosis but also another type of hyalohyphomycosis or phaeohyphomycosis (if melanized hyphae is observed). So it is not so easy to conclude based on microscopic direct examination that we are dealing with dermatophytosis.

Line 17

For complete clarity, I would add "However, the ITS sequencing alone cannot be used as a determinative tool in specific identification in case of some fungal taxa", what sounds more appropriate.

Line 25

As I know should be ,,...agents causing dermatophytoses ...”

Line 27

Should be ,,sequences for Arthrodermataceae...”

Lines 27-28

,,...suitable for the unambiguous identification of dermatophyte species.” sounds more appropriate.

Materials and Methods

Line 53

As I know should be ,,data not provided”

Line 67

Should be simply ,,1.5mM MgCl2” without ,,of”

Line 68

To standardize the way of writing measurement units, it should be "10 pM" instead of "10 picomoles"

Lines 72-73

,,All the products of the amplification reaction were visualized by electrophoresis...” sounds really better.

Line 73

As I suspect there should be ,,SYBR Green”

Line 74

,,UView transilluminator” sounds more professionally than ,,gel documentation system”

Results

Lines 93-94

Generic names of fungi should be written in italics. Please check the entire manuscript in this regard.

Line 109

Should be T. tonsurans

Line 118

Without dot after ,,significant”.

Line 129

To the best of my knowledge should be ,,T. verrucosum” instead of ,,T. verrucossum”

Line 132

There is something wrong ,,but us was observed” and as I suspect should be ,,but it was observed”

Line 133

,,...causative agent of dermatophytoses” is more appropriate.

Line 137

What about Arthroderma, namely A. uncinatum? This species was not included in this study?

Line 150

I would suggest changing the caption of figure 5 to a more accurate one, namely "Visualization of PCR products obtained for subtilisin region of studied fungi."

Line 160

To be more precise should be "NCBI nucleotide database"

Figure 1

In the case of figure 1, the resolution should be improved because the information in this form is unreadable. Similarly, I would suggest choosing other colors within the pie charts, namely more diverse ones. Species names should be written in italics.

Figure 2

In relation to figure 2, the resolution should be improved because the information in this form is unreadable. Similarly, I would suggest choosing other colors within the bar graphs, namely more diverse ones (The blue colors are almost indistinguishable). Shouldn't the posted bar graphs include standard deviations?

Figure 3, 4 and 5

There is a strange glow (background) visible within figure 3, making the figure looks like blurred. The resolution should be improved.

Figure 6

In the case of figure 6, the resolution should be improved because the information in this form is unreadable. Species names should be written in italics. Within the figure 6 you should put scale bar which indicates how many substitution per 10 nucleotide positions is there.

Discussion

Line 162

,,The identification of etiological agent...” sounds better than ,,The identification of causal agent...”

Line 167

The Authors quote literature no. 2, which does not relate to the issue described in this sentence at all. In my opinion, work no. 15 should be cited here or other as intended by Authors.

Line 178

According to the newest taxonomy of dermatophytes should be Trichophyton benhamiae instead of Arthroderma benhamiae

Line 183

As I know should be ,,Arthrodermataceae” instead of ,, Arthrodermataecae”

Lines 185-186

As I suspect should be ,,...causative agents of dermatophytoses in humans and animals.”

Line 192

As I suspect should be ,,unwanted sequences” (plural)

Lines 186-189

In Discussion, the Authors explain why, instead of the SUB1 gene for which the PCI is ~88%, they decided to choose the SUB7 gene for further research, for which the PCI is ~50%, but the SUB7 gene is also present in the genomes of T. mentagrophytes species complex. In my opinion, it is worth adding a short explanation for interested potential readers at this point in the discussion, namely what consequences this may have for diagnostics. Do you think, that expanding the spectrum of detectable dermatophyte species to include species of the T. mentagrophytes species complex (when selecting the SUB7 gene sequence) be at the expense of a decrease in the sensitivity of the method?

Line 202

Gene names should be written in italics and in capital letters. Please standardize their notation throughout the manuscript.

Line 203

As I know should be ,,in order to understand”

References

References require formatting and standardization of writing according to Author Guidelines for Manuscript Submissions. For example in ref. 7 the full names of the authors and only the first letter of the surname are given, while in ref. 5 for the same authors the full surname and the first letter of the name are provided.

Supporting Information

With reference to the table 2 in Supplementary Materials, Authors should explain the similarity between compared subtilisin subtypes in the legend below the table. Do the values presented here indicate the degree of homology between nucleotide sequences, %of identity, E value, Query cover, or something else? For potential reader it should be explained.

Reviewer #2: The paper presents an extensive comparative analysis was performed using a dataset of all available subtilisin gene sequences for Arthrodermatacea to evaluate if this gene would be suitable for the identification of dermatophyte species.

The idea is very good, but the work presents some details that suggest imprecision in the results: The sequences that you obtained were of few species. Why didn't you turn to other databases, or why didn't you amplify and sequence the gene of interest in a series of reference strains? It is important to consider that the taxonomy of dermatophytes has been changing and, possibly, the sequences used correspond to species that were identified based on the prevalent taxonomy of different times.

Lines 63-64: The authors indicate that primers were designed based on the complete subtilisin (SUB7) gene sequence of all Arthrodermataceae (KF146903, FJ348241, AY439111, AY437852)… but they only present the accession number of four species (KF146903, FJ348241, AY439111, AY437852): t menta, t tonsurans, t verrucosum y a. benhamiae… these are not all the Arthrodermataceae

Line 71: It is notable that the alignment temperature is very low (42°C). The authors should discuss how at such a low temperature and with short primers (16 nt) they achieve good specificity. These are not usually the parameters of a specific PCR.

Line 109: please correct T.interdigiatle

In abstract, improve the writing of the statement: The efficiency and limitations of the subtilisin gene as a diagnostic tool are currently limited.

Limitations of the study should be included in the discussion.

Please improve the visual quality of figures 1 and 2, mainly the text.

6. PLOS authors have the option to publish the peer review history of their article (what does this mean?). If published, this will include your full peer review and any attached files.

Reviewer #1: **Yes: **Mariusz Dyląg

Reviewer #2: No

---

## [Author Response · Author response to Decision Letter 0]

25 May 2024

RESPONSE TO REVIEWERS

EDITOR’S COMMENTS

Comment 1: In particular, the selected primer pairs need to be more extensively evaluated for their specificity and more strains need to be tested. The chosen alignment temperature is relatively low and the selected primer sequences show very high sequence similarity with alkaline serine proteases of Penicillium species. Are the authors sure that the amplified 200 bp with Penicillium DNA is unspecific? This could be checked by sequencing. I also reviewed the primer sequences with BLAST and found e.g. 100% sequence similarity of the FW primer with Penicillium verrucosum (XM_057218044) and Penicillium chrysogenum (AJ870492.2). Also the RW primer shows high sequence similarities with some Penicillium species. Therefore several Penicillium species should be tested by the authors. Does the amplified sequence always differ in size? Do the primers target the alkaline serein protease sequences of Penicillium species? These questions should be addressed in a revised manuscript.

Response: The query raised is highly relevant and the authors have performed additional tests towards addressing this issue.

i. The selected primer pair was designed using Trichophyton subtilisin sequences as described in the manuscript. Yes, the forward primer sequence shows sequence similarity with Penicillium verrucosum (XM_057218044) and Penicillium chrysogenum (AJ870492.2).However, the reverse sequence, though shows sequence similarities with the same sequences, the amplification stringency was targeted to subtilisin from Arthrodermatacae species and the amplicon size varies.

ii. Eight different Penicillium strains were tested in vivo (One of nine did not yield a product for subtilisin). There is amplification as presented in supplementary figure 1 but at ~300bp not at the targeted region of ~650bp. The PCR products were sequenced and all Penicillium products aligned with the uncharacterised protein of Penicillium citrinum(XM_056639834.1) and P. waksmanii uncharacterised protein (XM-057268380.1).Further, all subtilisin sequences were aligned sequences of the uncharacterised protein and alkaline serine proteases of Penciliium. The alkaline serine protease of Pencillium is very distinct from the subtilisins of Arthrodermataceae. Two sequences, Penicillium citrinum(XM_056639834.1) and P. waksmanii cluster with Trichophyton. However, on close examination, these sequences show <50% identity.

These points are addressed in the revised manuscript in the methodology (line 66), results (Line 169 to 176, Supplementary Figure 1 and 2 and Supplementary Sequence List 2) and discussion (Line 261-267).

Comment 2: Furthermore, the introduction needs some brief information about the subtilisin genes and the discussion has to include a comparison of the author's method with already established methods for the identification of Arthrodermataceae species. What are the pros and cons?

Response: In the revised manuscript, a brief introduction to subtilisin and its role in pathogenesis in dermatophytosis is included in the introduction. Line 21 to 31. 

 The revised manuscript includes a comparison of the subtilisin based method with already established methods for the identification of Arthrodermataceae species in the discussion. Line 199 to 208 and line 268-277.

 

REVIEWER #1

Comment 1: Introduction lacks basic information about subtilisin. It would be good to at least briefly mention what subtilisins are and what is their biological role. I would see here a few sentences like "Subtilisin-like proteases are a group of serine endoproteases, coded by seven genes (SUB1-7), which decompose keratin structures." It is worth emphasizing in your article why it is worth dealing with such proteins and encoding them genes. Such information will certainly be helpful to potential readers and will attract more interested researchers, and will also increase the attractiveness of your article. 

Response: The authors are grateful to the reviewers for raising this important topic.In the revised manuscript, a brief introduction to subtilisin and its role in pathogenesis in dermatophytosis is included in the introduction. Further emphasis on the significance of using such biomarkers is included in the discussion as well.

Line 21-31: 

Protein markers have better species identification power [11] but are not well-studied in fungi. For organisms infecting skin, like the dermatophytes, secreted proteases play a significant role in establishing infection and proliferating in the host [12]. Martinez et al [13] reported a comparative genome analysis of T. rubrum and other related dermatophytes to identify candidate genes involved in infection. One of the genes selectively present and expressed in the dermatophytes is the Peptidase S8, also known as subtilisin. Secretome analysis of T.behemiae and T.rubrum indicate upregulation of the endoproteinases known as subtilisins and fungalysins that are commonly encountered in the dermatophytes. These enzymes digest proteins into large peptides and subsequently break them down into short chain peptides and amino acids. Preliminary sulfitolysis of native keratin is required for the action of several proteolytic enzymes. However, subtilisins are capable of degrading native keratin [14]. Thus, it was hypothesized that the subtilisin gene can be targeted for the specific identification of agents causing dermatophytoses in humans as well as animals.

Comment 2: Additionally, in the "Discussion" the Authors should refer to other molecular methods used in the detection of dermatophyte species, such as matrix-assisted laser desorption/ionization time of flight mass spectrometry (MALDI-TOF MS) and nano-electrospray ionization mass spectrometry (nano -ESI-MS) to detect and identify the most frequently isolated dermatophytes in comparison with subtilisin gene used as protein marker. How and why would the solution based on sequences for subtilisins proposed by Authors be better?

Response: The revised manuscript includes a comparison of the subtilisin based method with already established methods for the identification of Arthrodermataceae species in the discussion.

Line 199-208

Molecular diagnostics platforms provide rapid detection and better sensitivity. Matrix-assisted laser desorption/ionization-time of flight mass spectrometry (MALDI-TOF) which scans for surface biomolecules is currently used for the rapid identification of a wide range of bacteria and yeast species. However, its application for dermatophyte detection is currently limited to the availability of custom databases [33,34,35]. Studies highlight the misidentification of Trichophyton species due to insufficient spectra database available for dermatophyte identification [36,37]. Molecular diagnosis for the dermatophytes is often by nucleic acid amplification techniques targeting the ITS and CHS1 regions [32,38,39, 40]. Several modifications of PCR like multiplex techniques [41,42,43], nested PCR [44], PCR-ELISA [45], restriction digestion of the PCR products [46,47] or real time detection [48,49] for these targets are reported.

Line 268-277

A major setback in culture dependent methods of fungal pathogens is the sensitivity and turnaround time. Rapid immunological test methods like ELISA and latex agglutination tests are proven sensitive for bacterial and viral infections, however, the sensitivity in fungal infections is low [76,77]. Despite enhanced sensitivity in terms of amount of template required in nucleic acid-based detection methods like qPCR, ITS and chitin targeting methods will continue to be non-specific due to their universal presence in all species of fungi. The extraction of fungal nucleic acid from the clinical specimen is an additional challenge. Non-nucleic acid-based identification technologies of spectral analysis like nano- electrospray ionization mass spectrometry to detect and differentiate the secreted proteins like subtilisin with turnover time within 10 minutes, with a requirement of less than 50 pmol of a substance and creation of superior databases for techniques like MALDI-TOF or applications in nanoparticle enhanced ELISA can revolutionize dermatophyte diagnosis.

Comment 3: All the figures and tables are appropriate for this type of article but require thorough correction. Figures in their current form are unacceptable. In general, the paper has a logical flow. The abstract well correspond with the main aspects of the work and the literature is properly selected.

As a reviewer I am obligated to pay attention even to less important weak points of this work and all mentioned below comments should be carefully considered.

Response: Figures have been revised and formatted as per the guidelines of the journal.

Comment 4: Title

I would suggest changing the title to a more appropriate one, namely: “The diagnostic potential of Peptidase S8 gene in the Arthrodermataceae causing dermatophytoses: A Metadata analysis”. Moreover, due to the fact that "Arthrodermataceae" are in the plural, I also propose changing ,,dermatophytosis” to "dermatophytoses"

Response: We appreciate the suggested change and propose the revised title as “The diagnostic potential of Peptidase S8 gene in the Arthrodermataceae causing dermatophytoses: A Metadata analysis”

Comment 5 : 

Abstract

Due to the repetition of the term "accurate", I suggest rephrasing the first sentence to more appropriate, namely "Unambiguous identification of dermatophytes is necessary for accurate clinical diagnosis and epidemiological implications".As I know should be ,,...of T.mentagrophytes species complex...”

In my opinion ,,...dermatophytes causing dermatophytoses..." sounds more correct.

To the best of my knowledge the genus Trichophyton split into three separate clades, namely: T. rubrum complex, T. benhamiae complex and T. mentagrophytes complex. The Authors did not mention anything about T. rubrum complex in the abstract. Can subtilisin be used for the differentiation of T. rubrum and what about PCI scores for several subtypes? In my opinion it will be significant for potential readers to know if differences within the subtilisin gene can further be used to differentiate members of the T. rubrum complex. However, if the Authors have not conducted such research, it is worth informing about it at least briefly. While all subtypes of subtilisin (gene variants) show low PCI scores and for this reason cannot be used for the differentiation of T. benhamiae species complex, SUB7 seems to be the best for T. mentagrophytes, but which one should be used for the differentiation of T. rubrum species complex? The Authors do mention it in "Results" (line 105), I quote "SUB1 showed good probability of identification for T. rubrum" but I think it is worth mentioning T. rubrum complex in the Abstract for full clarity.

Response: 

The abstract has been revised based on the constructive review received. Sentences are rephrased and reframed based on the inputs provided. Further, an overview of the current classification of the genera Trichophyton is provide and the ability and limitation of subtilisin to be used for the differentiation of T. rubrum and T.behemiae is included in the abstract, results and the discussion to provide better clarity.

Comment 6: 

Introduction

Line 2

As I know should be ,,... humans and animals...”

Line 3

I can`t agree that skin is ,,the non-living cornified layer”. Skin should be considered as an organ.

Line 6

The article by Kakande et all [2] refers to the prevalence of dermatophytoses in Uganda. Instead of this article I would like to suggest to cite more appropriate work in this field (for example: DOI:https://doi.org/10.1016/j.pathol.2016.08.006)

Line 9

To the best of my knowledge should be "While direct microscopic examination of skin scrapings indicates fungal infection, ...". Based on microscopic examination (10%KOH in DMSO or 5mM calcofluor white) we can observe blastoconidia, true hyphae and other fungal structures depending on fungal etiological agent of infection. If true hyphae is observed it may be dermatophytosis but also another type of hyalohyphomycosis or phaeohyphomycosis (if melanized hyphae is observed). So it is not so easy to conclude based on microscopic direct examination that we are dealing with dermatophytosis.

Line 17

For complete clarity, I would add "However, the ITS sequencing alone cannot be used as a determinative tool in specific identification in case of some fungal taxa", what sounds more appropriate.

Line 25

As I know should be ,,...agents causing dermatophytoses ...”

Line 27

Should be ,,sequences for Arthrodermataceae...”

Lines 27-28

,,...suitable for the unambiguous identification of dermatophyte species.” sounds more appropriate.

Response: 

We are grateful to the reviewers for the insights provided and accept the suggestions provided. The introduction has been modified based on the suggestions provided to emphasize the need for this study and provide better clarity on the concepts addressed.

Appropriate references have been cited to report global prevalence. 

Clarity on the scope and limitations of microscopic examination and ITS has been included in the discussion.

Line 192-214

Several microscopy dependent techniques play a vital role in the direct detection of fungal elements by wet mount preparation or stained slide preparation using bright field, phase contrast and fluorescence microscopy for the laboratory diagnosis of dermatophytosis. Though these techniques indicate a fungal infection, they are highly nonspecific [22,23,24,25,26,27]. The isolation and identification of etiological agent of dermatophytosis has long depended on culturing methods which stand as gold standard with the development of several modifications in SDA and potato dextrose agar with the inclusion of antibiotics widely used [28,29,30]. Major drawback of this gold standard is the low recovery of cultures from clinical specimen (<40%) and the long turn over time [31,32]. 

Molecular diagnostics platforms provide rapid detection and better sensitivity. Matrix-assisted laser desorption/ionization-time of flight mass spectrometry (MALDI-TOF) which scans for surface biomolecules is currently used for the rapid identification of a wide range of bacteria and yeast species. However, its application for dermatophyte detection is currently limited to the availability of custom databases [33,34,35]. Studies highlight the misidentification of Trichophyton species due to insufficient spectra database available for dermatophyte identification [36,37].

Molecular diagnosis for the dermatophytes is often by nucleic acid amplification techniques targeting the ITS and CHS1 regions [32,38,39, 40]. Several modifications of PCR like multiplex techniques [41,42,43], nested PCR [44], PCR-ELISA [45], restriction digestion of the PCR products [46,47] or real time detection [48,49] for these targets are reported. The use of ITS and chitinase based molecular detection methods have proven valuable [50]. However, the universal presence of these genes hinders specific detection and identification [51]. ITS region is successful in identifying organisms belonging to phylum Basidiomycota whereas for organisms belonging to phylum Ascomycota the success rate for the identification is reported lower [52]. Some organisms like Aspergillus have identical ITS region with other clinically important species [53]. The universal presence of ITS region in all eukaryotic organisms does not help differentiate dermatophytes from non-dermatophytes and within the dermatophytes. Thus, the correct identification of the dermatophytes remains disputed [17,54,55]. 

Comment 7: 

Materials and Methods

Line 53

As I know should be ,,data not provided”

Line 67

Should be simply ,,1.5mM MgCl2” without ,,of”

Line 68

To standardize the way of writing measurement units, it should be "10 pM" instead of "10 picomoles"

Lines 72-73

,,All the products of the amplification reaction were visualized by electrophoresis...” sounds really better.

Line 73

As I suspect there should be ,,SYBR Green”

Line 74

,,UView transilluminator” sounds more professionally than ,,gel documentation system”

Results

Lines 93-94

Generic names of fungi should be written in italics. Please check the entire manuscript in this

---

## [Decision Letter · Decision Letter 1]

12 Jun 2024

PONE-D-24-06282R1Diagnostic ability of Peptidase S8 gene in the Arthrodermataceae causing dermatophytoses: A Metadata analysisPLOS ONE

Dear Dr. Mohan Raj,

Thank you for submitting your manuscript to PLOS ONE. All critical points raised by the reviewers have been addressed, but there are still minor changes of the text and the figures needed (see comments of reviewer #1). Therefore, we invite you to submit a revised version of the manuscript that addresses the points raised during the review process.

We look forward to receiving your revised manuscript.

Kind regards,

Olaf Kniemeyer

Academic Editor

PLOS ONE

Journal Requirements:

Reviewers' comments:

Reviewer's Responses to Questions

**Comments to the Author**

1. If the authors have adequately addressed your comments raised in a previous round of review and you feel that this manuscript is now acceptable for publication, you may indicate that here to bypass the “Comments to the Author” section, enter your conflict of interest statement in the “Confidential to Editor” section, and submit your "Accept" recommendation.

Reviewer #1: All comments have been addressed

Reviewer #2: All comments have been addressed

2. Is the manuscript technically sound, and do the data support the conclusions?

Reviewer #1: Yes

Reviewer #2: Yes

3. Has the statistical analysis been performed appropriately and rigorously? 

Reviewer #1: Yes

Reviewer #2: Yes

4. Have the authors made all data underlying the findings in their manuscript fully available?

Reviewer #1: Yes

Reviewer #2: Yes

5. Is the manuscript presented in an intelligible fashion and written in standard English?

Reviewer #1: Yes

Reviewer #2: Yes

6. Review Comments to the Author

Reviewer #1: Review for manuscript assigned as: PONE-D-24-06282R1 (Manuscript Number)

Dear Authors,

Thank you so much for very accurate, conscientious and meticulous corrections and taking into account all the suggested changes. I really appreciate your effort. Your manuscript was significantly improved. Nevertheless, as a reviewer, I am obliged to point out some minor shortcomings in the manuscript that, in my opinion, still require/are worth improving.

Abstract

Lines 2-3

Instead of ,,...etiological agents of dermatophytosis...” should be ,,...etiological agents of dermatophytoses” (plural)

Introduction

Lines 7-9

To make it sound more professional and appropriate, I recommend rewording: ,, While based on direct microscopic examination of skin scrapings, the presence of fungal elements like conidia and true hyphae indicate fungal infections, the diagnosis is not conclusive and identification requires other culture-dependent or culture-independent methods.”

Results

Line 164

In case of ,,NCBI Nucleotide database” a capital letter in case of word ,,nucleotide” is not necessary.

Figure 1

The colors assigned in the legend to individual components of the pie chart are invisible, which makes it difficult for a potential reader to know which markings correspond to the data shown on the pie chart. The resolution of Figure 1 should also be improved.

Figure 2a

The colors assigned in the legend to individual components of the pie chart are invisible, which makes it difficult for a potential reader to know which markings correspond to the data shown on the pie chart. The resolution of Figure 2a should also be improved.

Figure 3 and 5

The resolution should also be improved.

Discussion

Line 232

To the best of my knowledge should be ,,...causative agents of dermatophytoses...” instead of ,,...causative agents of dermatophytosis...”

Reviewer #2: Thanks to the authors for responding to the suggestions, the presentation of the work improved considerably

7. PLOS authors have the option to publish the peer review history of their article (what does this mean?). If published, this will include your full peer review and any attached files.

Reviewer #1: **Yes: **Mariusz Dyląg

Reviewer #2: No

---

## [Author Response · Author response to Decision Letter 1]

22 Jun 2024

Response to reviewers

Reviewer #1

Comment 1

Abstract

Lines 2-3

Instead of ,,...etiological agents of dermatophytosis...” should be ,,...etiological agents of dermatophytoses” (plural) 

Response: We are grateful to the reviewer for the insights provided and accept the correction.

Revised manuscript Abstract Line 3 “...etiological agents of dermatophytoses”

Comment 2: 

Introduction

Lines 7-9

To make it sound more professional and appropriate, I recommend rewording: ,, While based on direct microscopic examination of skin scrapings, the presence of fungal elements like conidia and true hyphae indicate fungal infections, the diagnosis is not conclusive and identification requires other culture-dependent or culture-independent methods.”

Response: Sentence rephrased based on the inputs provided.

Comment 3: 

Results

Line 164

In case of ,,NCBI Nucleotide database” a capital letter in case of word ,,nucleotide” is not necessary.

Response: Typographical errors are corrected in the revised manuscript.

Comment 4 and 5: 

Figure 1 and Figure 2a

The colors assigned in the legend to individual components of the pie chart are invisible, which makes it difficult for a potential reader to know which markings correspond to the data shown on the pie chart. The resolution of Figure 1 should also be improved. 

Response: The colours assigned in the legend to individual components of the pie chart have been enlarged for better visibility. The resolution of all the images in the revised submission is 400 dpi .

Comment 6: 

Figure 3 and 5

The resolution should also be improved. 

Response: The resolution of all images in the revised submission is 400 dpi .

Comment 7:

Discussion

Line 232

To the best of my knowledge should be ,,...causative agents of dermatophytoses...” instead of ,,...causative agents of dermatophytosis...” 

Response: We are grateful to the reviewer for the insights provided and accept the correction. 

Revised manuscript Discussion Line 232 : “...causative agents of dermatophytosis...”

---

## [Editor Report · Decision Letter 2]

24 Jun 2024

Diagnostic ability of Peptidase S8 gene in the Arthrodermataceae causing dermatophytoses: A Metadata analysis

PONE-D-24-06282R2

Dear Dr. Mohan Raj,

We’re pleased to inform you that your manuscript has been judged scientifically suitable for publication and will be formally accepted for publication once it meets all outstanding technical requirements.

Kind regards,

Olaf Kniemeyer

Academic Editor

PLOS ONE
---

## [Editor Report · Acceptance letter]

28 Jun 2024

PONE-D-24-06282R2 

PLOS ONE

Dear Dr. Mohan Raj, 

I'm pleased to inform you that your manuscript has been deemed suitable for publication in PLOS ONE. Congratulations! Your manuscript is now being handed over to our production team.

Kind regards, 

on behalf of

Dr. Olaf Kniemeyer 

Academic Editor

PLOS ONE